# Immunoregulatory Property of C-Type Lectin-Like Receptors in Fibrosing Interstitial Lung Diseases

**DOI:** 10.3390/ijms21103665

**Published:** 2020-05-22

**Authors:** Wiwin Is Effendi, Tatsuya Nagano, Helmia Hasan, Resti Yudhawati

**Affiliations:** 1Division of Respiratory Medicine, Department of Internal Medicine, Kobe University Graduate School of Medicine, 7-5-1 Kusunoki-cho, Chuo-ku, Kobe 650-0017, Japan; wisepulmo@gmail.com; 2Department of Pulmonology and Respiratory Medicine, Medical Faculty of Airlangga University, Surabaya 60131, Indonesia; el_helmia@yahoo.com (H.H.); restiyudhawati@gmail.com (R.Y.)

**Keywords:** innate immunity, pattern recognition receptors, C-type lectin-like receptors, immunoregulatory receptor, tyrosine-based signaling motifs, interstitial lung diseases

## Abstract

The innate immune system identifies exogenous threats or endogenous stress through germline-encoded receptors called pattern recognition receptors (PRRs) that initiate consecutive downstream signaling pathways to control immune responses. However, the contribution of the immune system and inflammation to fibrosing interstitial lung diseases (ILD) remains poorly understood. Immunoreceptor tyrosine-based motif-bearing C-type lectin-like receptors (CTLRs) may interact with various immune cells during tissue injury and wound repair processes. Dectin-1 is a CTLR with dominant mechanisms manifested through its intracellular signaling cascades, which regulate fibrosis-promoting properties through gene transcription and cytokine activation. Additionally, immune impairment in ILD facilitates microbiome colonization; hence, Dectin-1 is the master protector in host pulmonary defense against fungal invasion. Recent progress in determining the signaling pathways that control the balance of fibrosis has implicated immunoreceptor tyrosine-based motif-bearing CTLRs as being involved, either directly or indirectly, in the pathogenesis of fibrosing ILD.

## 1. Introduction

Innate and adaptive immunity liaise with each other for defending the human body. The innate immune system is an evolutionarily conserved host defense system that encompasses virtually all tissues and may be triggered through germline-encoded receptors, including PRRs that are expressed both on immune and nonimmune cells [1]. These receptors recognize specific molecules known as pathogen-associated molecular patterns (PAMPs) and damage-associated molecular patterns (DAMPs), activating consecutive proinflammatory responses to eliminate infectious agents [2]. PRRs are divided into four significant subfamilies: Toll-like receptors (TLRs), nucleotide-binding oligomerization domain (NOD)-like receptors (NLRs), retinoic acid-inducible gene-1 (RIG-1)-like receptors (RLRs), and C-type lectin receptors (CLRs) [3].

CLRs constitute a superfamily of transmembrane proteins and soluble receptors that Ca^2+^-dependent carbohydrate-recognition domains (CRDs) recognize as self (endogenous) or nonself (exogenous) ligands [4]. Nevertheless, some CLRs, known as CTLRs, are also associated with CRDs and are not restricted to interacting with carbohydrates, such as β-glucan, binding many noncarbohydrate ligands, such as lipids and proteins [5]. CTLRs expressed by myeloid and nonmyeloid cells regulate intracellular signaling networks of the immune system and regulators of homeostasis via tyrosine-based signaling motifs [6]. A recent study showed that one cluster of the CTLR family, dendritic cell-associated C-type lectin-1 (Dectin-1), regulates fibrosis through macrophage colony-stimulating factor (M-CSF) expression [7]. In the balancing mechanism of fibrosis, CTLRs might be expressed in the initial phase of tissue injury and wound repair of the fibrosis process [8].

The mechanisms of immunoregulation are based on the recognition systems of antigen specificity and the interconnecting systems of innate immunity at different levels and manifest in various ways [9]. At the cellular level, Treg cells are essential for establishing and maintaining immunological self-tolerance and downregulating various immune responses to antigens of pathogens [10].

Intermolecular biochemical signaling events based on tyrosine phosphorylation are essential in balancing activating and inhibitory immune responses to microorganisms [11]. Immunoreceptor tyrosine-based activation motif (ITAM)- and immunoreceptor tyrosine-based inhibition motif (ITIM)-mediated signal transduction responses are needed for degranulation, phagocytosis, cell-mediated cytotoxicity, and the production of cytokines or various antimicrobial molecular species [12].

The interplay of signals transduced directly via tyrosine-based signaling motifs or indirectly via association with transmembrane adaptor proteins (TRAPs) demonstrates the plasticity and versatility of innate immunity [13]. In general, there are seven TRAPs, including linker for activation of T cells (LAT), protein-associated with GEMs (PAG)/carboxy (C)-terminal Src kinase (CSK)-binding protein (CBP), non-T-cell activation linker (NTAL)/linker for activation of B cells (LAB), LCK-interacting membrane protein (LIME), TCR-interacting molecule (TRIM), SH2-domain-containing protein tyrosine phosphatase (SHP2)-interacting TRAP (SIT) and linker for activation of X cells (LAX, where X denotes an unidentified cell) [14].

Focusing on tyrosine-based signaling motifs, in this review, we describe some CTLR families (Dectin-1 and MICL), discussing recent discoveries of their involvement in lung fibrotic mechanisms and their prospects as potential targets in ILD.

## 2. C-Type Lectin-Like Receptors

The characteristics of CTLRs are achieved when the second loop of normal CLRs (the extended loop region), which are Ca^2+^-dependent carbohydrates for binding ligands, is cleaved [15]. Myeloid CTLRs are essential molecules involved in a wide variety of cellular functions, such as inflammatory responses, endocytosis, phagocytosis, and cytokine production [16]. Indeed, in addition to serving as antigen-uptake receptors for internalization and presentation to T cells, CTLRs also trigger multiple signaling pathways that result in the activation of nucleus factor-κβ (NF-κβ), type I interferon (IFN), and/or inflammasomes, and mitogen-activating protein kinases (MAPK) [17,18].

Myeloid CTLRs, based on cytoplasmic signaling networks, can be divided into three leading groups: ITAM-bearing CTLRs, ITIM-bearing CTLRs, and non-immunoreceptor tyrosine-based motif-bearing CTLRs [18]. The signaling families of myeloid CTLRs are described in Figure 1.

### 2.1. ITAM- and HemITAM-Bearing CTLRs

The structure of complex immunoregulatory receptors consists of (1) the interface domain for target recognition at the cell surface and communication of extracellular stimuli to sophisticated intracellular signaling networks via (2) transmembrane (TM) segments and (3) cytoplasmic tail (CYT) regions of varying length [12]. The TM protein connects with at least one extracellular interface domain and one or more ITAMs within the CYT [19].

The classical motifs of ITAM are characterized by two intracellular YxxL/I sequences separated by 6 to 12 amino acids (YxxL/Ix(6-12)YxxL/I), where x represents any amino acid [20]. Initially, the ITAM was recognized only as a sequence in the CYT but was later found to be present in both the CYT of receptors and TRAPs, where it serves as a critical link to downstream signaling cascades [21].

The activation of ITAM-bearing CTLRs is associated with Syk or Zap-70, either directly as an integral component of a single ITAM-containing CYT (Dectin-1, CLEC-2, DNGR-1, and SIGN-R3) or with signaling TRAPs connected to classic ITAM motifs, such as the Fc receptor γ-chain (FcRγ) or DAP10/DAP12 (Dectin-2, hBDCA-2, mDCAR, mDCAR1, Mincle, and MDL-1) [22]. Upon activation, most CTLRs mediate intracellular signaling pathways associated with ITAM-bearing TRAPs [23].

Dectin-1 (CLEC7A, CLECSF12, CANDF4, CD369, and BGR) is one of the most intensively studied CTLRs with many of its characteristics well defined. Dectin-1 is a 28-KDa PRR that binds β-glucans via β-1,3 or both β-1,3 and β-1,6 glycosidic bonds [24] and an endogenous undefined ligand on T cells [25]. Dectin-1 is predominantly expressed on the surface of alveolar macrophages [26]. Human Dectin-1 has only N-linked glycosylation sites in the stalk region; however, N-linked glycosylation of Dectin-1 is essential for its recognition of fungal β-glucan and the subsequent activation of NF–κβ [27]. The “phagocytic synapse” mode allows Dectin-1-binding β-glucans on the cell wall of pathogens and soluble β-glucans, excluding the clustered receptors in tyrosine phosphatases CD45 and CD148, which are activated by particulate β-glucans [28]. These synapses ensure that phagocytosis and the production of reactive oxygen species (ROS) occur only when an invading microorganism is physically close [29].

β-glucans are natural polysaccharides produced by bacteria, yeast, fungi, and many plants. The innate immune system recognizes β-glucans in their various forms and translates recognition into intracellular signaling, stimulates cellular responses, and participates in orchestrating the adaptive immune response [30]. β-glucans act as PAMPs and are recognized by PRRs on innate immune cells since they cannot directly penetrate the cell membrane due to their large molecular size [31]. Indeed, β-glucans were involved in the pathogenesis of sarcoidosis [32], fungal asthma allergy [33], and idiopathic pulmonary fibrosis (IPF) [34]. 

Similar to other type II transmembrane CTLRs, Dectin-1 has a single extracellular C-type lectin-like domain (CTLD) connected to single-pass TM by an amino acid stalk region and a single ITAM/hemITAM-containing CYT [35]. In addition, Dectin-1 also contains a triacidic domain (DED) for triggering phagocytosis [36]. A “DED” motif sequence precedes the YxxL sequence in both Dectin-1 and CLEC-2 [37].

Upon ligand binding to the CRD, tyrosine residues are phosphorylated, presumably via Src-family kinases (Lyn and/or Fyn), and Syk or Zap-70 kinases are rapidly recruited and activated via their SH2 domains [38]. In contrast to traditional ITAMs, only one tyrosine within a single YxxL motif is sufficient to mediate an interaction with Syk [39]. How does Dectin-1 bind Syk with a single “hemITAM”? Specific residues may stabilize the interaction between Dectin-1 and Syk within the CYT, or Dectin-1 may dimerize or form higher-order complexes once bound by a ligand [40]. In addition, Syk is more readily activated by autophosphorylation than is Zap-70 [41].

Syk is a 72-KDa nonreceptor tyrosine kinase that consists of two N-terminal Src homology 2 (SH2) motif (SH2-N) and and C-terminal SH2 motif (SH2-C) domains separated by two interdomains: A and B [42]. Syk is activated upon binding of the tandem SH2 domains to the ITAM, and an increase in Syk catalytic activity results in autophosphorylation of the tyrosine residues in interdomain B to create docking sites for interactions with other signaling molecules [43]. Active Syk is directly associated with members of the VAV and phospholipase Cγ (PLCγ) families, the p85α subunit of phosphoinositide 3-kinase (PI3K), and SH2 domain-containing leukocyte protein 76 (SLP76) and SLP65 [44].

Dectin-1 can switch on two signaling cascades, the Syk-dependent and Syk-independent pathways, via Raf-1 to induce immune responses [45]. The activation of NF-κβ is a significant outcome [46]. The canonical pathway initiated via activation of Syk and subsequent phosphorylation of PLCγ2 enables the activation of the caspase-recruitment domain family 9 (CARD9)-B cell lymphoma 10 (BCL10)-mucosa-associated lymphoid tissue lymphoma translocation gene 1 MALT1/CARD9-BCL10-MALT1 (CBM) complex via PKCδ. The CBM complex then activates Iκβ kinase (IKK), releasing NF-κβ subunits p65 and c-Rel:p50, which are translocated into the nucleus, where they regulate gene transcription [18]. Nevertheless, CARD9 is useless for antigen receptor signaling based on Carma1 as a link to Bcl10–Malt1 [47]. Additionally, Dectin-1 stimulates a noncanonical pathway utilizing NF-κβ-inducing kinase (NIK)-dependent, but CARD9-independent, activation of the NF-κβ subunit RelB:p52 [48].

Dectin-1 signaling activates dendritic cells (DCs), converting Treg into Th17 and IL-17 responses [49]. The Dectin-1/Syk pathway, but independently of TLRs, can also bridge innate immunity to the adaptive immune system via CD4+ T-helper cells, B cells, and CD8+ cytotoxic T cells [50]. Only TLR-4 cooperates with Dectin-1 to induce adaptive Th17 immunity against fungal infection [51].

The Syk-independent pathway of Dectin-1 is mediated through the DED involved with Raf-1 activation, leading to the phosphorylation and permitting the subsequent acetylation of the NF-κβ p65 subunit or convergence with noncanonical pathways at the level of NF-κβ [18]. Raf-1 activation enhances the expression of some Syk-dependent cytokines, including IL-10, IL-12 p35, IL-12/23 p40, IL-6, and IL-1β, but negatively regulates the RELB-dependent cytokine IL-23 p19 [22]. In addition, Dectin-1 signaling through the Raf-1 pathway induces immunological memory via functional reprogramming to protect against reinfection [52].

In addition to its role in the NF-κB pathway, Dectin-1 signaling results in the activation of p38, extracellular signal-regulated kinase (ERK), and JUN-N/amino-terminal kinase (JNK) cascades. CARD9 activates Dectin-1-induced ERK activation by linking Ras-GRF1 to H-Ras [53] and stimulates the receptors Nod2-associated p38 and JNK23 [54]. Additionally, Dectin-1-triggered nuclear factor of activated T cells (NFAT)/calcineurin depends on Syk activation of PLCγ followed by soluble IP3 binding with endoplasmic Ca^2+^ channels, leading to the dephosphorylation and translocation of NFAT into the nucleus of T cells [18]. NFAT activation in macrophages and DCs regulates the induction of early growth response (EGR) 2 and EGR 3 transcription factors and the key inflammatory mediators cyclooxygenase-2 (COX-2), interleukin-2 (IL-2), IL-10, and IL-12p70 [55]. The mechanisms of the signaling cascade of Dectin-1 are described in Figure 2.

### 2.2. ITIM-Bearing CTLRs

The first ITIM was recognized in a family of low-affinity Fc receptors for immunoglobulin G (IgG) (FcγRIIB) that coaggregated with BCR, inhibiting mouse B-cell activation; however, FcγRIIB was also shown to inhibit TCR-dependent T-cell and FcR-dependent mast cell activation [56]. These observations make clear that the termination of immune responses is not merely due to the loss of activating signals; hence, the paradigm focuses on how the coligation of ITAM- and ITIM-containing coreceptors results in the attenuation or inhibition of cellular response [57]. The ITIM present in the CYT with the prototype 6–amino acid ITIM sequence is (Ile/Val/Leu/Ser)-X-Tyr-X-X-(Leu/Val), where X denotes any amino acid [58]. Recent evidence indicates that, under specific conditions, incomplete phosphorylation of ITAM/ITIM may paradoxically lead to the transduction of negative (ITAMi) and positive (ITIMa) signaling [59].

Upon the coaggregation with activating receptors, tyrosine phosphorylation of ITIMs provides docking sites for SH2 domain-containing inositol phosphatase (SHIP-1 and SHIP-2) or SH2 domain-containing protein tyrosine phosphatase (SHP-1 and SHP-2) [60]. SHIP contains a phosphatase domain, an N-terminal SH2 domain, a pleckstrin homology-related (PH-R) domain, and a C-terminus [61]. In addition, SHP is a nonreceptor protein tyrosine phosphatase (PTP) that has an N-terminal SH2, followed by a catalytic domain/PTP domain and a C-SH2 terminus tail [62].

SHIP recruitment to the ITIM of FcγRIIB is a dominant mechanism controlling SHIP activity and B cell activation via the PI3K pathway [63]. The PI3K pathway is a target substrate for SHIP through the hydrolyzation of the membrane phosphoinositide PI(3,4,5)P3 to PI(3,4)P2, whereas SHP-1 directly removes the phosphate groups on tyrosine [64].

Both SHPs modulate cellular signaling that involves PI3K, Akt, Janus kinase 2 (JAK2), signal transducer and activator of transcription proteins (STAT), and MAPK [65]. Interestingly, SHPs are involved in several cellular activities, including the regulation of cell growth [66] and MAPK activity [67].

One of the myeloid inhibitory CTLRs is MICL, which can inhibit cellular activation mediated by other immunoreceptors. However, intermolecular interactions between immunoregulatory receptor families or receptor crosstalk among ITIM-bearing CTLRs may not always reflect the simple recruitment of phosphatases and the inhibition of Syk signaling [22].

MICL (DCAL-2). Human MICL is expressed in most myeloid cells, such as granulocytes, monocytes, macrophages, and DCs, but not lymphocytes [68]. It has three isoforms: α, β, and γ, with the β-isoform lacking a transmembrane domain and γ-isoforms lacking the CTLD [18]. Human MICL is highly N-glycosylated in primary cells with various levels of glycosylation of other cell types, but the receptors appear to be expressed as monomers [69]. The specific ligands for MICL are not well defined; however, MCL might bind trehalose dimycolate (TDM) [70].

MICL contains a single ITIM in its CYT that can associate with SHP-1 and SHP-2, but not SHIPs, to modulate cellular activity [71]. MICL induces ERK activation, and its coengagement with TLR4 suppresses IL-12 expression and the polarization of naive T cells into Th1 cells [72]. However, the effect of SHP1 and SHP2 on ERK activation does not always lead to increase activity; therefore, it remains to be determined whether this is a downstream pathway of MICL [73]. The signaling networks of MICL are described in Figure 3.

## 3. Immune Mechanisms in Fibrosing Interstitial Lung Diseases

ILD encompasses a large and heterogeneous group of parenchymal lung disorders, which may be related to systemic diseases or environmental exposure and unknown causes [74]. One of the most common types of ILD is IPF, a progressive fibrosing ILD that is characterized by a decline in lung function and predominantly affects older adults with a histologic pattern of usual interstitial pneumonia (UIP) [75].

Patients with certain other types of chronic fibrosing ILD are also at risk of developing a progressive phenotype, such as idiopathic nonspecific interstitial pneumonia (NSIP), unclassifiable idiopathic interstitial pneumonia, autoimmune ILDs, sarcoidosis, chronic hypersensitivity pneumonitis (HP), asbestosis or silicosis [76].

The pathogenesis of IPF is unclear; however, chronic and/or repetitive microinjuries of the alveolar epithelium are considered triggers of the disease [77]. Injury from toxic particles or infectious agents triggers inflammatory responses leading to an imbalance of cytokines, increases in free radicals, release of growth factors, and the development of lung fibrosis [78].

Dysregulated activation of Wnt/β-catenin signaling, inflammasomes, CC chemokine ligand 18 (CCL18), M1/M2 macrophage polarization, IL-17, IL-6, IL-1β, and transforming growth factor-beta (TGF-β) are immune cells and mediators involved in IPF [79]. Additionally, a dominant mechanism through which innate immune cells adopt fibrosis-promoting properties through PRRs is implicated in a rapidly progressive form of IPF [80].

ILD tends to impair immunity; thus, patients have a higher risk of fungal colonization, which plays a prominent role in the deterioration in the condition of people with the disease [81]. A recent study from China showed that patients with connective tissue diseases (CTDs) that have underlying interstitial pneumonia and who were previously treated with prednisone, or multiple antibiotics, were more likely to develop invasive pulmonary fungal infections (IPFIs) [82]. However, Dectin-1 is a key player in IL-17 production in the pulmonary defense against *Aspergillus fumigatus* [83,84].

Compared to other PRRs, particularly TLRs, the involvement of CTLRs in ILD is enigmatic. Nevertheless, the role of immunoregulatory CTLRs in regulating inflammation and homeostasis demonstrates that these receptors perhaps modulate inflammation-associated lung diseases.

## 4. Immunoregulatory Receptors as A Novel Therapeutic Targets for ILD

Corticosteroids are the recommended drug for either mono- or combination therapy with other immune modulators, especially for patients with acute exacerbation of the disease [85]. Currently, clinical trials with novel drugs, mainly antifibrosis, anti-cytokine, and immunoregulatory drugs, are being investigated in various trial phases [86]. Dectin-1, acting directly and indirectly, is involved with the pathogenesis of ILD due to its role in regulating gene transcription, including genes of the cytokines and chemokines that are involved in inflammation and fibrosis.

The proinflammatory cytokine interleukin-17 (IL-17) is necessarily implicated in several chronic inflammatory diseases that often culminate in organ damage followed by impaired wound healing and fibrosis development [87]. DCs are converted into hybrid Th cells producing IL-17 upon Dectin-1 stimulation [49]. Furthermore, IL-17 has roles in the acute exacerbation of idiopathic pulmonary fibrosis via a Dectin-1-dependent mechanism [88] and the development of fibrosis in experimental HP [89,90].

Dectin-1 is also involved in regulating the susceptibility of a patient with cystic fibrosis (CF) [91,92,93] and asthma [94]. Expression of CXCR4^+^-associated Dectin-1 is prominent in fungal cystic fibrosis lung disease; therefore, it may serve as a potential biomarker and therapeutic target in fungal cystic fibrosis lung disease.

The high level of CCL18 in patients with fibrotic lung diseases is indicative of pulmonary fibrotic activity [95] and is a potential biomarker for IPF [96] and for the early identification of progressive ILD [97]. Moreover, high concentrations of Dectin-1-associated IL-10 and CCL18 induce the polarization shift of macrophages toward an alternatively activated macrophage (M2) phenotype in lung fibrosis [98,99]. A recent study revealed that Dectin-1 may inhibit the secretion of CCL18 and may be valuable as a therapeutic agent to prevent M2 macrophage polarization and progressive fibrotic lung disease [100].

In line with the role of Dectin-1 in innate immunity and homeostasis, the Syk signaling pathway is involved in the mechanism of parenchymal lung fibrosis. Multiple pathways of protein-tyrosine kinases, such as Syk, Src, and Fyn, are involved in promoting fibroblast proliferation and matrix production [101] and play critical roles in the pathogenesis of pulmonary fibrosis [102]. The Syk inhibitor fostamatinib prevents bleomycin-induced fibrosis and inflammation in the skin and lung by reducing Syk phosphorylation and TGF-β expression [103].

Intriguingly, Dectin-1, via interactions with Syk and the Nod-like receptor protein 3 (NLRP3) inflammasome, is associated with the activation of NF–κβ, which then induces proinflammatory cytokines such as IL-6, TNF-α, and pro-IL-1β [104]. Syk associates directly with apoptosis-associated speck-like protein (ASC) and NLRP3 but interacts indirectly with procaspase-1; specifically, Syk phosphorylates ASC at Y146 and Y187 residues to enhance ASC oligomerization and the recruitment of procaspase-1 [105]. NLRP3 is essential in the progression of pulmonary fibrosis [106]. Activation of the NLRP3 inflammasome induces caspase-1 cleavage and then leads to the secretion of the profibrotic mediators IL-1β and IL-18 [107].

In contrast to Dectin-1, MICL is involved indirectly through protein intracellular cascades in fibrosis. As previously noted, SHP is involved in the PI3K/Akt pathway. The activation of PI3K/Akt signaling can modulate β-catenin-mediated Wnt signaling via the inhibition of β-catenin nuclear localization and regulation of glycogen synthase kinase 3β (GSK3β) phosphorylation [108,109]. Recently, SHP-1 was found to regulate pulmonary fibrosis via the inhibition of β-catenin in lung epithelial cells [110]. Additionally, mice deficient in SHP-1 were found to more susceptible to bleomycin-induced lung injury and fibrosis [111]. However, we assume that PI3K/Akt-dependent Wnt/β-catenin is not directly related to MICL.

Wnt/β-catenin plays a critical role in development and adult tissue homeostasis, especially during abnormal wound repair and fibrogenesis [112]. Additionally, the interaction of TGF-β and the canonical Wnt/β-catenin pathway stimulates fibroblast accumulation and myofibroblasts [113,114] and the development of pulmonary fibrosis [115]. Therefore, blocking Wnt/β-catenin signaling attenuates myofibroblast differentiation of lung resident mesenchymal stem cells and pulmonary fibrosis [116] and bleomycin-induced pulmonary fibrosis [117].

MAPK-phosphatase (MKP)-5 is required for the induction of changes to lung fibroblasts and bleomycin-induced lung fibrosis [118]. Activation of the MAPK/ERK signaling cascade in the lungs in human fibrosis is associated with fibrogenesis, including fibroblast growth, proliferation, and survival [119].

## 5. Conclusions

Both Dectin-1 and Syk have essential roles in NLRP3/NFAT-associated fibrosis via caspase-1, which induces the transformation of IL-1β from pro-IL-1β. Also, Dectin-1 upregulates the production of IL-17 and TGF-β. The discovery that the β-glucan ligand binds to Dectin-1 in M2 macrophages has been a breakthrough in the use of Dectin-1 as a potential biomarker and targeted treatment in fibrosing ILD.

In contrast to Dectin-1, MICL indirectly affects the mechanism of fibrosis via intracellular signaling proteins, such as SHP, Wnt/β-catenin, and PI3K/Akt. Even though the mechanism remains unclear, the role of immunoregulatory receptors of CTLRs in fibrosing ILD is relevant and exciting, particularly for its clinical implications. Additional studies are required to shed light on the role of Dectin-1 and especially MICL in the pathogenesis of fibrosing ILD.

## Figures and Tables

**Figure 1 ijms-21-03665-f001:**
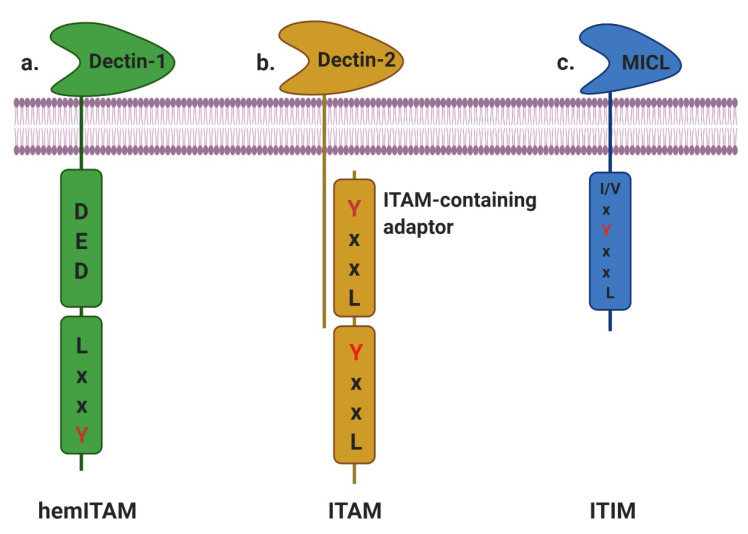
Schematic representation of myeloid CTLRs based on cytoplasmic signaling networks and the binding of early adaptors. (**a**) hemITAM-bearing CTLRs transduce signals via Syk through a single tyrosine-based motif in their tail; Dectin-1. (**b**) ITAM-bearing CTLRs transduce signals via Syk through association with ITAM-bearing adaptors such as FcRγ chain or DAP-12; Dectin-2. (**c**) ITIM-bearing CTLRs possess an ITIM motif; MICL.

**Figure 2 ijms-21-03665-f002:**
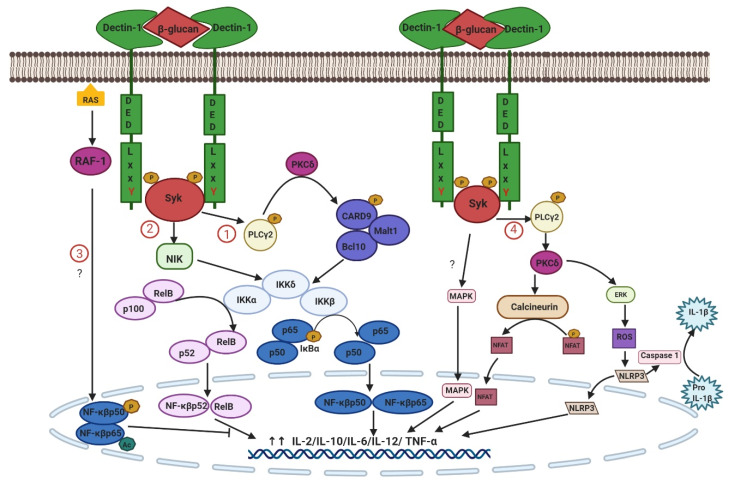
Signaling cascade of Dectin-1. The binding of β-glucan on the CRD induces phosphorylation of YxxL (hemITAM), which in turn initiates the recruitment of Syk to the two phosphorylated receptors and other cascades. (**1**) The canonical Syk-dependent pathway. Active Syk phosphorylates PLCγ2, enabling the subsequent activation of the CBM complex via PKCδ. The CBM complex removes its inhibitor, IκB, through activation of the IKK complex and stimulates all NF-κB subunits. (**2**) The noncanonical Syk-dependent pathway. Dectin-1 utilizes NIK-dependent but CARD9-independent activation of the NF-κB subunit RelB: p52, leading to the activation of NF-κB RelB and p52. The Raf-1-mediated/Sky-independent pathway can abrogate this pathway. (**3**) The Syk-independent pathway mediated through DED involves Raf-1 and the activation of NF-κB p65. (**4**) Activation of NLRP3, NFAT/calcineurin, and ERK/JUN via Dectin-1-ky depends on PLCγ2; Caspase 1 induces pro-IL-1β to IL-1β. The activation of NF-κB, NLRP3, NFAT, ERK/JUN, and MAPK results in the modification of gene transcription, including that of cytokines and chemokines.

**Figure 3 ijms-21-03665-f003:**
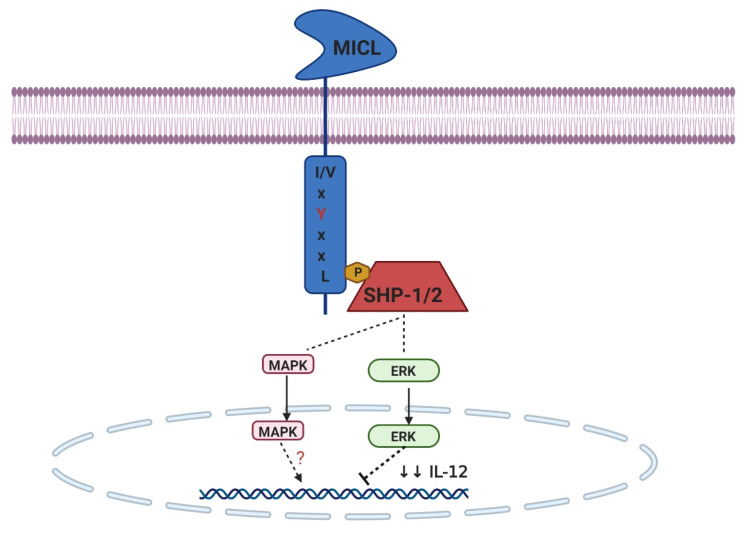
The signaling networks of MICL. The ITIM motif within MICL interacts with SHP-1 and SHP-2. MICL inhibits cytokine transcription instigated by other receptors. ERK may suppress IL-12 expression and the polarization of Th1 cells, whereas another signaling induces their activation through MAPK. Dashed arrows represent a pathway that has yet to be fully defined.

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
