# Peer review of "Immunoregulatory Property of C-Type Lectin-Like Receptors in Fibrosing Interstitial Lung Diseases"

_ijms, 2020, doi:10.3390/ijms21103665_

Round 1
Reviewer 1 Report
In this review study Authors reported recent findings concerning the involvement of Dectin-1 and MICL signalling , components of C-type lectin receptors (CLRs) families, in fibrotic process observed in interstitial lung diseases. The prospects of such components of immune systems as potential target were also addressed.
As underlined by Authors the role of immune systems and inflammation in the pathological mechanisms that lead to fibrosing interstitial lung disease is poorly understood. I think that the concept is of great interest and particularly important for its clinical implications.
Author Response
Response to Reviewer 1 Comments
Point 1:
In this review study Authors reported recent findings concerning the involvement of Dectin-1 and MICL signalling, components of C-type lectin receptors (CLRs) families, in fibrotic process observed in interstitial lung diseases. The prospects of such components of immune systems as potential target were also addressed.
As underlined by Authors the role of immune systems and inflammation in the pathological mechanisms that lead to fibrosing interstitial lung disease is poorly understood. I think that the concept is of great interest and particularly important for its clinical implications.
Response 1:
I emphasize the role of CTLRs in fibrosing ILD although its mechanism is poorly understood
Page 9, line 321-323.
Even though the mechanism remains unclear, the concept of immunoregulatory receptors of CTLRs in fibrosing interstitial lung disease is relevant and exciting, particularly for its clinical implications.

Reviewer 2 Report
The manuscript by Wivendi et al., disccusses the recent advances in determining the signaling pathways that control the balance of fibrosis focusing on immunoreceptor tyrosine-based motif-bearing CTLRs. CTLRs have benn implicated both either directly or indirectly, in the pathogenesis of fibrosing interstitial lung diseases (ILD). The manuscript is well organized and links some new and interesting aspects of sihnaling interactions to the pathogenesis of ILD.
I beleive that some points could be further developed.
-DEctin-1 has been shown to affect aspects of both innate immunity and lymphocyte activation and possibly to present an important link between innate and acquired immunity.
-The importance of glycan contribution to CTLRs signaling patways needs to be further highlighted.
Author Response
Response to Reviewer 2 Comments
The manuscript by Wivendi et al., disccusses the recent advances in determining the signaling pathways that control the balance of fibrosis focusing on immunoreceptor tyrosine-based motif-bearing CTLRs. CTLRs have benn implicated both either directly or indirectly, in the pathogenesis of fibrosing interstitial lung diseases (ILD). The manuscript is well organized and links some new and interesting aspects of sihnaling interactions to the pathogenesis of ILD.
I beleive that some points could be further developed.
Point 1:
-DEctin-1 has been shown to affect aspects of both innate immunity and lymphocyte activation and possibly to present an important link between innate and acquired immunity.
Response 1:
I add information about the important role of Decin-1 to bridge innate and acquired immunity.
Page 4, line 150-153.
Dectin-1 signaling activates dendritic cells (DCs), convert Treg into Th17 and IL-17 responses [49]. Also, the Dectin-1/Syk pathway but independently TLRs can bridge innate immune to the adaptive immune system via CD4+ T-helper cells, B cells, and CD8+ cytotoxic T cells [50]. Only TLR-4 cooperates with Dectin-1 to induce adaptive Th17 immunity against fungal infection [51].
Point 2:
-The importance of glycan contribution to CTLRs signaling patways needs to be further highlighted. 

Response 2:
I describe the role of b-glucan in PRRs/CTLRs signaling pathways and specifically in ILD.
Page 4, line 112-118.
b-glucans are natural polysaccharides produced by bacteria, yeast, fungi, and many plants. The innate immune system recognizes b-glucans in its various form and translates recognition into intracellular signaling, stimulates cellular responses, and participates in orchestrating the adaptive immune response [30]. b-glucans act as PAMPs and recognized by PRRs on innate immune cells since they cannot directly penetrate the cell membrane due to their large molecular size [31]. Indeed, b-glucans were involved in the pathogenesis of sarcoidosis [32], fungal asthma allergy [33], and idiopathic pulmonary fibrosis (IPF) [34]
